# Bi-Level Dynamic Parameter Sharing among Individuals and Teams for Promoting Collaborations in Multi-Agent Reinforcement Learning

## Abstract

Parameter sharing has greatly contributed to the success of multi-agent reinforcement learning in recent years. However, most existing parameter sharing mechanisms are static, and parameters are indiscriminately shared among individuals, ignoring the dynamic environments and different roles of multiple agents. In addition, although a single-level selective parameter sharing mechanism can promote the diversity of strategies, it is hard to establish complementary and cooperative relationships between agents. To address these issues, we propose a bi-level dynamic parameter sharing mechanism among individuals and teams for promoting effective collaborations (BDPS). Specifically, at the individual level, we define virtual dynamic roles based on the long-term cumulative advantages of agents and share parameters among agents in the same role. At the team level, we combine agents of different virtual roles and share parameters of agents in the same group. Through the joint efforts of these two levels, we achieve a dynamic balance between the individuality and commonality of agents, enabling agents to learn more complex and complementary collaborative relationships. We evaluate BDPS on a challenging set of StarCraft II micromanagement tasks. The experimental results show that our method outperforms the current state-of-the-art baselines, and we demonstrate the reliability of our proposed structure through ablation experiments.

## 1 Introduction

In many areas, collaborative Multi-Agent Reinforcement Learning (MARL) has broad application prospects, such as robots cluster control (Buşoniu et al., 2010), multi-vehicle auto-driving (Bhalla et al., 2020), and shop scheduling (Jiménez, 2012). In a multi-agent environment, an agent should observe the environment's dynamics and understand the learning policies of other agents to form good collaborations. Real-world scenarios usually have a large number of agents with different identities or capabilities, which puts forward higher requirements for collaborations among agents. Therefore, how to solve the large-scale MARL problem and promote to form stable and complementary cooperation among agents with different identities and capabilities are particularly important.

To solve the large-scale agents issue, we can find that many collaborative MARL works adopting the centralized training paradigm use the full static parameter sharing mechanism (Gupta et al., 2017), which allows agents to share parameters of agents' policy networks, thus simplifying the algorithm structure and improving performance efficiency. This mechanism is effective because agents generally receive similar observation information in the existing narrow and simple multi-agent environments. In our Google Research Football (GRF) (Kurach et al., 2020) experiments, we can find that blindly applying the full parameter sharing mechanism does not improve the performance of algorithms because the observation information is very different due to the movement of different players. At the same time, because the full static parameter sharing mechanism ignores the identities and abilities of different agents, it constantly limits the diversity of agents' behavior policies (Li et al., 2021; Yang et al., 2022), which makes it difficult to promote complementarity and reliable cooperation between agents in complex scenarios.

Recently, in order to eliminate the disadvantage of full parameter sharing, a single-layer selective parameter sharing mechanism has been proposed (Christianos et al., 2021; Wang et al., 2022), that is, extracting deep features from agents' observation information by an encoder and clustering them to achieve the combination of different agents in order to select different agents for parameter sharing.

Although the single-level selective parameter sharing mechanism can promote the diversity of agents' policies, it causes the relationship between agents that do not share parameters simultaneously fragmented. So that agents cannot establish complementary cooperative relationships in a broader range. More importantly, designing an effective selector is the key to sharing selective parameters, especially for the single-level dynamic selective parameter sharing mechanism, which needs to be done within several rounds of the selection operation. Most methods only use the real-time observation information of agents, which loses attention to agents' history and is not conducive to correctly mining the implicit identity characteristics of the agents. For a football team, special training for different players, such as shooting training for the forwards and defense training for the defenders, will be carried out. However, winning a game requires special training and coordination between players of different roles. That is, not only do we need to share parameters with agents of the same role, but we also need to combine agents of different identities to ensure that they can form robust and complementary collaboration on a larger scale.

To address these issues, in this paper, we propose a bi-level dynamic parameter sharing mechanism among individuals and teams (BDPS). The advantage functions of agents can be expressed as the advantages of taking action relative to the average in the current state. We consider that the advantage function can better represent the actual roles of agents in current identity and grouping. In order to more accurately identify the role of agents, at the individual level, we compute the long-term advantage information of the agents as the key to virtual role identification and use the variational autoencoders (VAE) (Kingma & Welling, 2014) to learn the distribution of the role characteristics of the agents, and obtain more accurate virtual role directly by sampling the distribution of role features. To alleviate the split of the relationship between different virtual role agents by the single-layer dynamic parameter sharing mechanism, we further use the graph attention network (GAT) (Velickovic et al., 2018) to learn the topological relationships between different roles based on the roles obtained at the individual level, so as to achieve the combination of different identity agents at a higher level and a broader range. Through the method we designed, we achieve dynamic and selective parameter sharing for agents at two different levels, individual and team, and achieve the goal of stabilizing complementary collaboration among agents in a more extensive scope while achieving the diversity of agents' policies.

We test BDPS and algorithms using different parameter sharing mechanisms on the StarCraft II micromanagement environments (SMAC) (Samvelyan et al., 2019) and the Google Research Football (GRF) (Kurach et al., 2020). The experimental results show that our method not only outperforms other methods with single-level selective parameter sharing mechanism and full parameter sharing mechanism in general on all super hard maps and four hard maps of SMAC, but also performs well in the experimental scenarios in the used GRF. In addition, we carried out ablation experiments to verify the influence of different parameter sharing settings on the formation of complementary cooperation between agents, which fully proves the reliability of our proposed method.

## 2 BACKGROUND

### 2.1 DECENTRALIZED PARTIALLY OBSERVABLE MARKOV PROCESS

A full cooperative MARL task can usually be modeled as a decentralized partially observable markov process (Dec-POMDP) (Oliehoek & Amato, 2016). We can define a tuple $\mathcal{M} = \langle \mathcal{N}, \mathcal{S}, \mathcal{A}, R, P, \Omega, O, \gamma \rangle$ to represent it, where $\mathcal{N}$ is the finite set of $n$ agents, $s \in \mathcal{S}$ is a finite set of global state, and $\gamma \in [0, 1)$. At each time step $t$, each agent $i \in \mathcal{N}$ receives a local observation $o_i \in \Omega$ according to the observation function $O(s, i)$, takes an action $a_i \in \mathcal{A}$ to form a joint action $\boldsymbol{a} \in \mathcal{A}^n$, and gets a shared global reward $r = R(s, \boldsymbol{a})$. Due to the limitation of partial observability, each agent conditions its policy $\pi_i(a_i|\tau_i)$ on its own local action-observation history $\tau_i \in \mathcal{T} \equiv (\Omega \times \mathcal{A})^*$. Agents together aim to maximize the expected return, that is, to find a joint policy $\boldsymbol{\pi} = \langle \pi_1, ..., \pi_n \rangle$ to maximize a joint action-value function $Q^{\boldsymbol{\pi}} = \mathbb{E}_{s_{0:\infty}, \boldsymbol{a}_{0:\infty}} \left[ \sum_{t=0}^{\infty} \gamma^t r_t | s_0 = s, \boldsymbol{a}_0 = \boldsymbol{a}, \boldsymbol{\pi} \right]$.

## 2.2 CENTRALIZED TRAINING WITH DECENTRALIZED EXECUTION

In many MARL settings, partial observability problems can be solved by centralized training with decentralized execution (CTDE) paradigm (Lowe et al., 2017; Foerster et al., 2018; Rashid et al., 2020; Wang et al., 2021a; Son et al., 2019), which is currently the mainstream of MARL methods. The centralized mode is adopted in training, and after the training, agents only make decisions based on their local observation and the trained policy network. Thus, the problems of unstable environment and large-scale agents can be overcome at the same time to a certain extent.

## 2.3 VARIATIONAL AUTOENCODER

Variational autoencoder (VAE) (Kingma & Welling, 2014) is a generative network model based on Variational Bayesian (VB) (Fox & Roberts, 2012) inference. Two probability density distribution models are established by the inference network $q_\phi$ and the generative network $p_\xi$. The inference network is used for the variational inference of the original input data $x$ to generate the variational probability distribution of hidden variables $z$. The generative network restores the approximate probability distribution of the original data according to the generated implicit variable variational probability distribution. VAE model can be divided into the following two processes: the approximate inference process of posterior distribution of hidden variables: $q_\phi(z|x)$ and the generation process of conditional distribution of generated variables: $p_\xi(z) p_\xi(\hat{x}|z)$.

In order to make $q_\phi(z|x)$ and the true posterior distribution $p_\xi(z|x)$ approximately equal, VAE uses KL-divergence to measure the similarity between them:

$$D_{KL}(q_\phi(z|x) \| p_\xi(z|x)) = \log p_\xi(x) + E_{q_\phi(z|x)} [\log q_\phi(z|x) - \log p_\xi(x,z)], \quad (1)$$

where the term $\log p_\xi(x)$ is called log-evidence and it is constant. Another term is the negative evidence lower bound (ELBO). The VAE with additional GRU network is widely used in sequence anomaly detection (Su et al., 2019). In this paper, we use it to identify the identity feature distribution mapped behind the advantage information sequence of agents for a period of time to better guide them in choosing appropriate parameter sharing partners.

## 2.4 GRAPH ATTENTION NETWORK

Graph attention network (GAT) (Velickovic et al., 2018) is a network architecture based on an attention mechanism, which can learn different weights for different neighbors through the attention mechanism. For calculating the output characteristics of node $i$, GAT first trains a shared weight matrix $\mathbf{W}$ for all nodes to obtain the weight of each neighbor node of node $i$. Then, according to the weight, the attention coefficients between node $i$ and its neighbor nodes are calculated. Finally, these attention coefficients are weighted and summed to obtain the output features $\boldsymbol{h}'_i$ of node $i$:

$$\boldsymbol{h}'_i = \sigma_1 \left( \sum_{j \in \mathcal{N}_i} \frac{\exp\left(\sigma_2\left(\boldsymbol{a}^T[\mathbf{W}\boldsymbol{h}_i \| \boldsymbol{h}_j]\right)\right)}{\sum_{k \in \mathcal{N}_i} \exp\left(\sigma_2\left(\boldsymbol{a}^T[\mathbf{W}\boldsymbol{h}_i \| \boldsymbol{h}_k]\right)\right)} \mathbf{W}\boldsymbol{h}_j \right)_{(1-\text{Head})}, \quad (2)$$

where $\sigma_1$ and $\sigma_2$ represent nonlinear functions, $\mathcal{N}_i$ represents the first-order neighborhood of node $i$, $\boldsymbol{a}^T$ represents the transpose of the weight vector $\boldsymbol{a}$ and $\|$ is the concatenation operation.

In recent years, MARL with communication has widely used GAT to determine the explicit communication goals of agents (Niu et al., 2021; Seraj et al., 2021). In this paper, we use the GAT to establish the topological relationship between different virtual roles, and further combine the agents under different virtual role mappings to form complementary and reliable cooperative relationships.

## 3 OUR METHOD

In this section, we introduce the proposed BDPS in detail. BDPS mainly comprises individuals and teams, as shown in Figure 1. We get inspiration from people: a person may play multiple roles and can freely switch roles in different scenes, which promotes the stable development of people, and the same is true for agents. Therefore, we choose to sacrifice some network parameters for agents to maintain multiple roles and groups at the same time. And according to the role selector $U$ and the group selector $V$, choose the most suitable role and group for the agents.

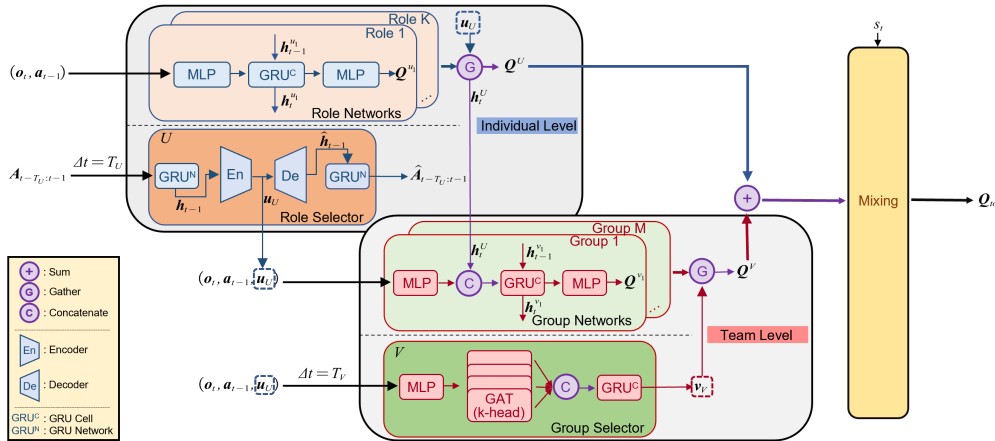

Figure 1: The overall BDPS architecture.

First, we explain the relationship between agents and individual virtual roles and groups.

**Definition 1** *Given n (n≥2) agents, k (2≤k≤n) roles and m (2≤m≤k) groups, we have the following mapping relationships: $f_U : \mathcal{N} \mapsto \mathcal{K}$ and $f_V : \mathcal{K} \mapsto \mathcal{M}$, where $\mathcal{K}$ and $\mathcal{M}$ represent finite sets of $k$ roles and $m$ groups, respectively.*

*From the $f_V$, we can see the agents' groups depend on agents' roles, so we give the time relationship for agents to update the groups and roles: $T_V = e \cdot T_U$, where $T_V$ represents the update period of the group, $T_U$ represents the identity's update period and $e \in Z^+$.*

## 3.1 INDIVIDUAL LEVEL DESIGN

Different from the existing methods of determining the roles of agents based on real-time observation information, we consider the impact of the long-term cumulative advantages of agents on the definition of the agents' roles, because the advantage information can better represent the current state of agents than the observation information. At the same time, unlike the existing methods that only use the encoder to learn the role characteristics of agents for clustering, we learn the long-term advantages of agents through the VAE to obtain the distribution information of role. Because in many cases, the role of an agent will not change just because it makes a unique action. At the individual level in Figure 1, we first maintain the role selector $U$ to select appropriate roles $\boldsymbol{u}_U$ for agents, then apply parameter sharing to agents according to the same role.

### 3.1.1 CHOOSE APPROPRIATE ROLES FOR AGENTS

Firstly, the role selector $U$ inputs the pre-calculated agents' advantage information sequence $\boldsymbol{A}_{t-T_U:t-1} = \{\boldsymbol{A}_{t-T_U}, \boldsymbol{A}_{t-T_U+1}, ..., \boldsymbol{A}_{t-1}\}$ into a GRU network to capture complex temporal dependence between agents' advantage information in the virtual roles' update period. Secondly, when the time meets the periodic condition of updating virtual roles, we take the implicit output information $\boldsymbol{h}_{t-1}$ at this time as the characteristics of the agents and input it into the Encoder $E_\phi$ of the VAE. Finally, we obtain the virtual roles $\boldsymbol{u}_U$ of agents from the $k$-dimensional Gaussian distribution output by the Encoder $E_\phi$.

For agent $i$, its advantage information at time $t$ can be obtained by $\boldsymbol{A}_t^i = \max\{\boldsymbol{Q}_t^i\} - \boldsymbol{Q}_t^i$, where $\boldsymbol{Q}_t^i$ represents the agent $i$'s local $Q$-function at time $t$, and unlike the classical calculating advantage function $\boldsymbol{A} = \boldsymbol{Q} - \boldsymbol{V}$, ours $\boldsymbol{A}_t^i \geq 0$ is convenient for us to use the Decoder $D_\xi$ to reconstruct the input information. When the virtual role $\boldsymbol{u}_U^i$ of agent $i$ needs to update at time $t$, its virtual role can be obtained from the VAE's bottleneck:

$$\boldsymbol{u}_U^i = \arg\max\{\boldsymbol{z}_U^i\}, \tag{3}$$

where $\boldsymbol{z}_U = \boldsymbol{\mu}_U + \boldsymbol{\sigma}_U \cdot \boldsymbol{\epsilon}_U, \boldsymbol{\epsilon}_U \sim \mathcal{N}(0, \boldsymbol{I})$.

## 3.2 Team Level Design

The purpose of introducing grouping is to eliminate the fragmentation of the relationship between agents caused by the single-layer parameter sharing mechanism, and to establish the cooperative relationship between agents of different roles in a wider range. As shown in Figure 1, the team level is mainly composed of the group selector $V$ and grouping cooperative networks. The group selector $V$ is realized through an additional reinforcement learning task. We introduce a GAT for this task to find the correlation between different dynamic roles and promote the role composition in the grouping results to be more comprehensive and complete. In the part of parameter sharing, we add additional implicit information $\boldsymbol{h}_t^U$ from the individual role level to give full play to the positive impact of the individual identities on the team results.

### 3.2.1 Combine Different Roles to Achieve Grouping

The group selector $V$ takes the virtual roles $\boldsymbol{u}_U$ as inputs. This input information is encoded into the feature vector of the agents via a multi-layer perceptron (MLP). Subsequently, these feature vectors form a set $\left\{\boldsymbol{x}_t^1, \boldsymbol{x}_t^2, ..., \boldsymbol{x}_t^N\right\}$ to be input into the GAT. We use the self-attention mechanism in the GAT to calculate the attention coefficient $e_{ij}$ between agents and use it as an essential basis for grouping:

$$e_{ij} = \text{LeakyReLU}\left(\boldsymbol{a}^T\left[\mathbf{W}\boldsymbol{x}_t^i||\mathbf{W}\boldsymbol{x}_t^j\right]\right), \tag{4}$$

where $i, j \in \mathcal{N}$. In our method, considering the similar observation information and close location arrangement between agents, and there is no rigid separation restriction between roles of different agents, we allow the agent's attention to come from all undead agents (including itself) at the current moment. Of course, to facilitate the comparison of the attention coefficient between agents, we also use the softmax function for normalization.

$$\boldsymbol{x}_t^{i'} = \overset{K}{\underset{k=1}{||}} \sigma\left(\sum_{j \in \mathcal{N}_i} \alpha_{ij}^k \mathbf{W}^k \boldsymbol{x}_t^j\right), \tag{5}$$

where $k$ represents the number of attention heads, which is consistent with the number of virtual roles we set. We hope to find the influence of different agents' roles as the dominant factors on the formation of teams by agents. For other parts in Equation 5, $\mathcal{N}_i$ represents the first-order neighborhood of agent $i$, and $\alpha_{ij} = \text{softmax}_j\left(e_{ij}\right)$ represents the normalized attention coefficient that indicates the importance of agent $j$'s features to agent $i$.

A new set of feature vectors $\boldsymbol{x}_t' = \left\{\boldsymbol{x}_t^{1'}, \boldsymbol{x}_t^{2'}, ..., \boldsymbol{x}_t^{N'}\right\}$ is obtained after the GAT, which is input into the GRU network to get the hidden state affecting the grouping of agents. When the time $t$ increment is equal to the period for agents to update groups $T_V$, we use the hidden state to output the groups' value and use the $\varepsilon-greedy$ to select groups $\boldsymbol{v}_V$ for agents.

## 3.3 Overall Objectives

### 3.3.1 Get the Local $Q$-Functions Required by Agents

In previous sections, we use the role selector $U$ at the individual level and the group selector $V$ at the team level to map agents from actual individuals to virtual roles $\boldsymbol{u}_U$ and groups $\boldsymbol{v}_V$, respectively. The bi-level parameter sharing mechanism is also established for agents using roles and groups information obtained. Although contributions from hidden states at the individual level are used in calculating the local $Q$-functions of agents at the team level, we still may not completely discard explicit efforts at the individual level of agents. So we give the local $Q$-function:

$$\boldsymbol{Q}\left(\boldsymbol{\tau}, \boldsymbol{a}\right) = \boldsymbol{Q}^U\left(\boldsymbol{\tau}, \boldsymbol{a}\right) + \boldsymbol{Q}^V\left(\boldsymbol{\tau}, \boldsymbol{a}\right). \tag{6}$$

### 3.3.2 Train the Modules Included in Our Method

Starting from the individual level, we need to train the role selector $U$ to select appropriate virtual roles for agents. The training objectives of the Role Selector $U$ include the corrected reconstruction item and the KL-divergence item, and our goal is to minimize these two items:

$$\mathcal{L}_U\left(\xi, \phi; \boldsymbol{A}_U\right) = \mathcal{L}_{\text{mse}}\left(\boldsymbol{A}_U, \hat{\boldsymbol{A}}_U\right) + \lambda D_{KL}\left[q_\phi\left(\boldsymbol{z}_{l_U}|\boldsymbol{h}_{l_U}\right)||p_\xi\left(\boldsymbol{z}_{l_U}\right)\right], \tag{7}$$

where $\mathcal{L}_{\mathrm{mse}}(\cdot)$ is the mean squared error term for calculate the reconstruction loss of the advantage sequence, $l_U$ is the length of the $\boldsymbol{A}_U$, $\lambda$ is a scaling factor and $\hat{\boldsymbol{A}}_U$ is the agents' advantage information reconstruction sequence, which is obtained by the Decoder's output $\hat{\boldsymbol{h}}_{l_U}$.

For the group selector $V$ at the team level, we used the QMIX (Rashid et al., 2018) and RODE (Wang et al., 2021b) methods because we introduced an additional deep reinforcement learning task to generate groups for agents:

$$\mathcal{L}_V(\theta_\nu) = \sum_b \left[ \left( \sum_{\Delta t = 0}^{T_V - 1} r_{t + \Delta t} + \gamma \max_{\boldsymbol{v}'_V} \bar{\boldsymbol{Q}}_{tot}^V(\boldsymbol{\tau}', \boldsymbol{a}', \boldsymbol{u}'_U, s') - \boldsymbol{Q}_{tot}^V(\boldsymbol{\tau}, \boldsymbol{a}, \boldsymbol{u}_U, s) \right)^2 \right], \quad (8)$$

where $b$ is the batch size of transitions sampled from the replay buffer.

In order for agents to use global rewards to learn local $Q$-functions, we input the local $Q$-value into the mixing network of QMIX (Rashid et al., 2018) again to estimate the global action value $\boldsymbol{Q}_{tot}(\boldsymbol{\tau}, \boldsymbol{a})$:

$$\mathcal{L}_{TD}(\theta_\mu) = \sum_b \left[ \left( r + \gamma \max_{\boldsymbol{a}'} \bar{\boldsymbol{Q}}_{tot}(\boldsymbol{\tau}', \boldsymbol{a}', s') - \boldsymbol{Q}_{tot}(\boldsymbol{\tau}, \boldsymbol{a}, s) \right)^2 \right]. \quad (9)$$

## 4 EXPERIMENTS

In this section, we demonstrate and evaluate the advantages of our proposed BDPS using the challenging tasks in the StarCraft II micromanagement enviroments (SMAC) and Google Research Football (GRF). We not only compare our proposed method with QMIX (Rashid et al., 2018) which adopts full parameter sharing mechanism and no parameter sharing mechanism, but also further compare with several baseline methods aimed at promoting the diversity of agents' policies, such as CDS (Li et al., 2021), EOI (Jiang & Lu, 2021) and RODE(Wang et al., 2021b). Finally, we ablate our method in SMAC, verifying the true utility of the components in our method.

### 4.1 PERFORMANCE ON GOOGLE RESEARCH FOOTBALL (DEC-POMDP)

In the official multi-agent example, agents are allowed to observe all information on the field, which is contrary to our research hypothesis. We provide an observation space setting for agents in the Dec-POMDP version. The observation space of agents can dynamically change with their motion vectors. See Appendix A.2 for details.

First, we compare our method and baseline methods in the GRF of the Dec-POMDP version we provide. Compared with the experimental scenario provided by CDS (Li et al., 2021), we use the algorithm to control all agents in the left team, use the built-in *scoring* and *checkpoints* reward settings of GRF, and set that the agents in the left team cannot be moved by the built-in AI of the system when they do not touch the ball.

As shown in Figure 2, we select two scenarios *academy_3_vs_1_with_keeper* and *academy_pass_and_shoot_with_keeper* for comparison. We compare the actual reward received by the agents with the goal score in both scenarios. In general, our method is able to achieve better results than other baseline algorithms in both scenarios. Among them, the full parameter sharing version of QMIX is better than the non-sharing version, and other improved baseline algorithms based on full parameter sharing QMIX can also achieve goals, but the effect is not obvious.

As we mentioned in Section 1, the full parameter sharing mechanism ignores the identities and capabilities of agents, which makes them lose the diversity of behavior policies. As a result, full parameter sharing cannot improve the effectiveness of the algorithm when agents observe scenes with large differences in information. For the single-layer dynamic selective parameter sharing, due to the hard cutting of the relationship between agents that do not participate in parameter sharing, although better results can be achieved by relying on the selective parameter sharing mechanism, the agents cannot form a sufficiently stable cooperative relationship, which makes the effect fluctuate significantly. See Appendix C for the specific results.

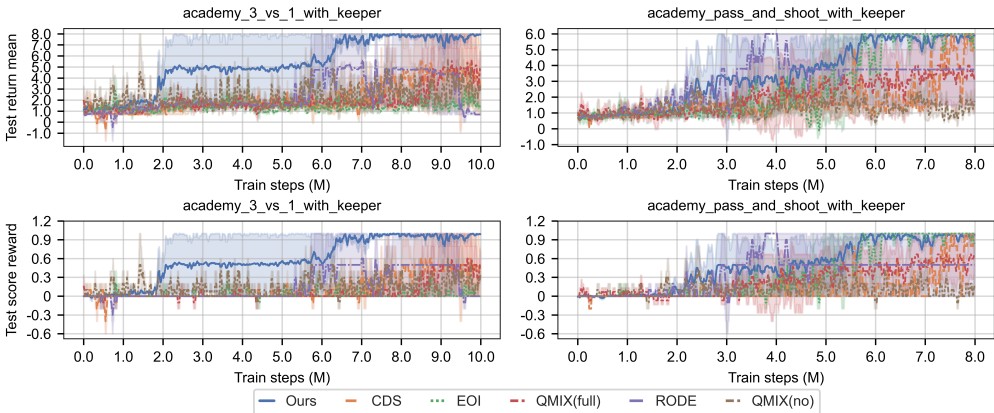

Figure 2: Performance comparison with baselines on Google Research Football.

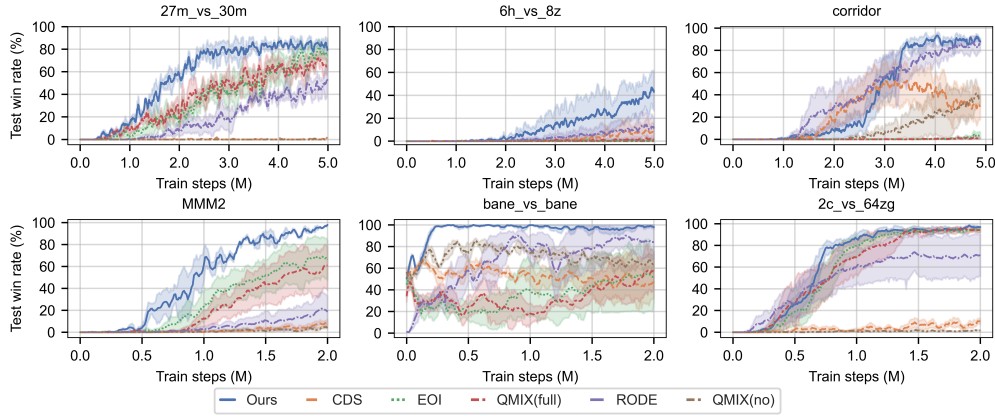

Figure 3: Performance comparison with baselines on three super hard maps and three hard maps. (In Appdendix E, we show results on the whole benchmark.)

In both scenarios, we can see that RODE differentiated the roles of the agents by limiting their action space, but this limitation also prevented the agents from achieving further training results. Through the experimental results, we can find that these baseline algorithms drop the ball in these two *academy*-scenarios, indicating that the agents do not form a sufficiently complementary collaborative relationship.

## 4.2 PERFORMANCE ON STARCRAFT II

As can be seen from Figure 3, our method is superior to other baseline methods on these maps, especially in maps with a larger number of agents, where our method is able to maintain its advantage. Of course, EOI and RODE also show good stability in these maps, and the full parameter sharing version of QMIX is still better than the no parameter sharing version.

Specifically, our method has more advantages than the CDS, which emphasizes the balance of personalities and commonalities of agents. As we mentioned, exploring the dynamic balance between individual personalities and commonalities is essential. Still, this balance is difficult to define, so we do not give factors in Equation 6 that can adjust the different importance at the individual and team levels. This balance is dynamic and hides in the process of sharing dynamic parameters we design. We will describe it in detail in the ablation experiment in Section 4.3.

## 4.3 ABLATION STUDY

In this section, we conduct ablation studies to understand the actual utility of each level in our bi-level dynamic parameter sharing mechanism. In addition to showing the winning rate of maps in SMAC, we calculate the entropy difference between individuals and teams to quantify the advantages of different components.

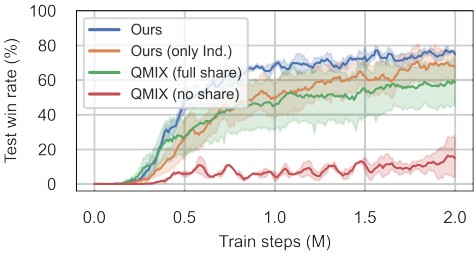
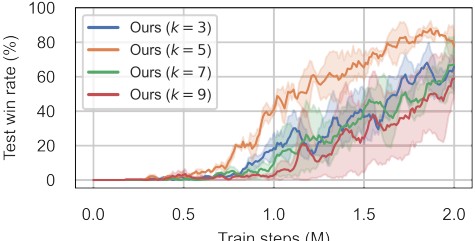

Figure 4: Comparison of different parameter sharing mechanisms in 5m_vs_6m.

Figure 5: Comparison of different virtual roles in MMM2 (Only individual level).

We first use the homogeneous agents' map 5m_vs_6m in SMAC to conduct ablation research to analyze the advantages of dynamic parameter sharing over full and no parameter sharing. As shown in Figure 4, we can see that the method of applying parameter sharing has apparent advantages over the method without parameter sharing. Of course, our method is better than QMIX's full static parameter sharing. Specifically, the single-level dynamic parameter sharing has a weak advantage over the full static parameter sharing. Our bi-level dynamic parameter sharing is more advantageous than the single-level dynamic and full static, which is exactly the goal of our proposed method.

As shown in Figure 5, we also compare the impact of different numbers of virtual roles on the collaboration of agents in the heterogeneous agents' map MMM2. From the curve in the figure, we can find that the winning rate in the MMM2 map increases first and then decreases with the number of virtual roles, which indicates that blindly increasing the number of virtual roles can adversely affect the collaboration of agents.

To quantify the difference between our single-level and bi-level dynamic parameter sharing, we consider analyzing from the perspective of information theory. First, we introduce the evaluation indicator: Entropy difference.

Entropy describes uncertainty. We describe the capabilities of agents in different layers by comparing the entropy of action value between the two layers.

$$\Delta H = \frac{1}{N} \sum_{i=1}^{N} \left( \sum_{a_i} p_V^i \cdot \log \frac{1}{p_V^i} - \sum_{a_i} p_U^i \cdot \log \frac{1}{p_U^i} \right), \tag{10}$$

where, $p_V^i = \text{softmax} \boldsymbol{Q}_V^i \left( \tau^i, a_{t-1}^i, u_U^i \right)$ and $p_U^i = \text{softmax} \boldsymbol{Q}_U^i \left( \tau^i, a_{t-1}^i \right)$.

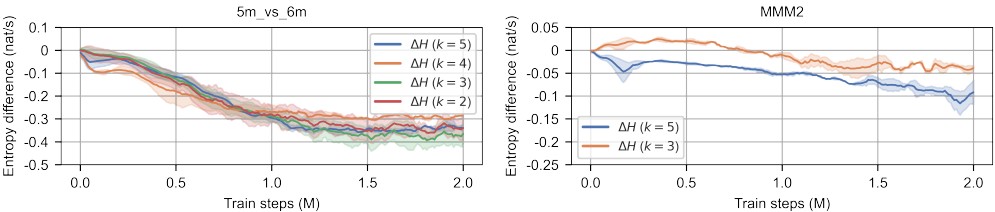

Figure 6: Comparison of the entropy difference of two levels in 5m_vs_6m and MMM2.

As shown in Figure 6, we can see that in 5m_vs_6m, the information entropy at the team level is always smaller than that at the individual level, and the difference between the two is increasing. The change of entropy shows that the agents at the team level are more orderly than strategies

formed by agents only at the individual level, which also confirms that our team-level design can find commonalities among agents with different virtual roles to establish correct collaborations between agents. For MMM2 in Figure 6, our conclusion is still valid. No matter whether $k = 3$ or $k = 5$, the entropy difference between the team level and the individual level is constantly expanding with the training, and the entropy of the team level is constantly developing in a direction smaller than that of the individual level. And this phenomenon is consistent with our winning rate in Figure 4 and Figure 5.

## 5 RELATED WORK

### 5.1 PARAMETER SHARING

Parameter sharing plays an important role in MARL. Tan (1993) first studied the positive role of "sharing" in promoting collaborative agents in classical reinforcement learning algorithms. Gupta et al. (2017) proposed a parameter sharing variant of the single agent DRL algorithms, which introduced the parameter sharing mechanism into homogeneous MARL. Obviously, parameter sharing has been widely used as the implementation details of homogeneous multi-agent algorithms, such as QMIX (Rashid et al., 2018), QTRAN (Son et al., 2019), Qatten (Yang et al., 2020), etc. Recently, Terry et al. (2020) applied parameter sharing extensions to heterogeneous agents algorithm by padding based method, demonstrating again the important use of parameter sharing for multi-agent algorithms. Of course, recent works have pointed out that full parameter sharing mechanism tend to make the behavior strategies of agents more same, Christianos et al. (2021) proposed a selective parameter sharing mechanism to eliminate the limitations of full parameter sharing.

### 5.2 INDIVIDUALS AND TEAMS

Focusing on the expected development of individuals and teams will not only help agents to maintain a variety of policies but also help them form a more stable collaboration. At the individual level, Wang et al. (2020; 2021b) focus on discovering the character traits behind the agents. Jiang & Lu (2021) studies the identifiability of agent trajectories and fixed identities. Du et al. (2019) proposes the use of internal rewards to stimulate diverse behaviors among agents. At the team level, Iqbal et al. (2021) randomly groups agents into related and unrelated groups, allowing agents to explore only specific entities in their environment. Wang et al. (2022) implements a dynamic grouping method by extracting the potential intentions of agents as tags. Li et al. (2021) divides agents' value functions into shared and unshared parts to focus on the agents' personality and commonality. In this paper, we focus on the impact of both individual and team levels on the complementary collaboration of agents. Of course, our approach is not to overlap the two layers of individuals and teams but to fully extend and combine the individual identities acquired by the agents.

## 6 CONCLUSION AND FUTURE WORK

In this paper, we proposed BDPS, a novel bi-level dynamic parameter sharing mechanism in MARL. By maintaining a role selector and a group selector, BDPS provides a new solution for agents to select the partners for parameter sharing in a timely and dynamic manner at both individual and team levels. And we integrate the roles and groups of agents into a whole to achieve a dynamic balance between their personalities and commonalities. Our experiments on SMAC and GRF show that BDPS can significantly facilitate the formation of complementary and reliable collaboration between agents.

Additionally, although we defined that agents have the opportunity to be in any role or grouping, they cannot explicitly utilize past knowledge when they were in other roles or groups. Therefore, how to make explicit use of the role knowledge they have learned and how to make use of other similar knowledge are issues that we need to explore further. We will present this paradigm of knowledge flow across roles, groups, and agents in future work.

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

# A MULTI-AGENT ENVIRONMENTS

In this paper, we use three multi-agent environments, Starcraft II Multi-agent Challenge (SMAC)(Samvelyan et al., 2019), Google Research Football (GRF) (Kurach et al., 2020) and Level-Based Foraging (LBF) Papoudakis et al. (2021); Albrecht & Ramamoorthy (2015); Albrecht & Stone (2019), to conduct verification experiments.

## A.1 SMAC

The starcraft multi-agent challenge is a fully collaborative, partially observable set of multiagent tasks. This environment implements various micro-management tasks based on the popular real-time strategy game StarCraft II. Each mission is a specific battle scenario in which a group of agents, each of whom controls a single unit, fight against an army controlled by the built-in AI in StarCraft II.

Compared with the LBF, SMAC provides more abundant battle scenes for agents. We conducted comparative experiments with baseline methods on hard and super hard maps. The characteristics of relevant maps are shown in Table 1.

Table 1: Maps used in our experiments.

| Map Name | Number of Controlled Agents | Map Type | | |
|---|---|---|---|---|
| bane_vs_bane | 24 | Hard | Symmetric | Heterogeneous |
| MMM2 | 10 | Hard | Asymmetric | Heterogeneous |
| 5m_vs_6m | 5 | Hard | Asymmetric | Homogeneous |
| 3s_vs_5z | 3 | Hard | Asymmetric | Homogeneous |
| 2c_vs_64zg | 2 | Hard | Asymmetric | Homogeneous |
| 27m_vs_30m | 27 | Super Hard | Asymmetric | Homogeneous |
| 6h_vs_8z | 6 | Super Hard | Asymmetric | Homogeneous |
| corridor | 6 | Super Hard | Asymmetric | Homogeneous |
| 3s5z_vs_3s6z | 8 | Super Hard | Asymmetric | Heterogeneous |

## A.2 GRF

In Google Research Football (GRF) tasks, agents need to cooperate according to the rules of football matches to win games or related training tasks. The reward settings of GRF can be divided into two types. One is that only goals can be rewarded ($scoring$), and the other is that when an agent reaches a specific coordinate point ($checkpoints$), it will get a certain reward. Compared with the two types of reward, the setting of the first type is more sparse. In the official multi-agent example, agents are allowed to observe all information on the field, which is contrary to our research hypothesis. We provide an observation space setting for agents in the Dec-POMDP version. The observation space of agents can dynamically change with their motion vectors, as shown in Figure 7. Considering the size of the field in the environment, we set the visual distance of the agents to $0.84$, and the forward visual angle to $200°$. The agents can observe the position and direction of the ball, and within the range of viewing them, we also allow the agents to observe the position of their teammates and the position of their opponents.

In our experiment, we control all the players on the left side and set their movement state to be lazy (when not touching the ball, there is no built-in AI intervention), and the built-in AI in the system completely controls all the players on the right side.

## A.3 LBF

The LBF is a multi-agent collection task built on a grid world where each agent and object is assigned a level. The criteria for successful collection by agents is the sum of the levels of the related agent needs to be equal to or greater than the level of the item. The agent will receive a reward at the relative item level if the item is successfully collected. LBF provides a more sparse reward environment and collaboration constraints than SMAC, which puts forward higher requirements for the effectiveness of multi-agent algorithms.

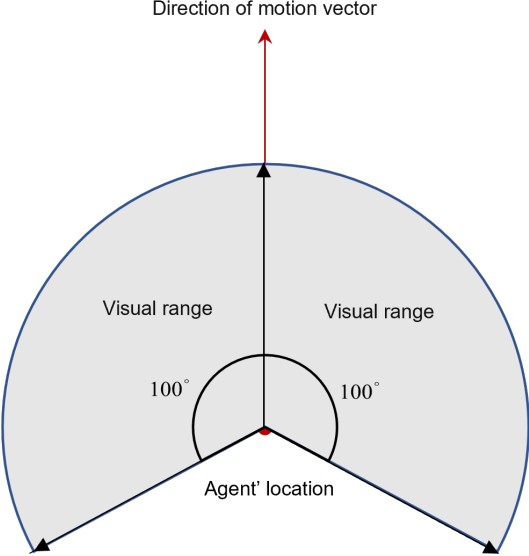

Figure 7: The Observation Space Design of Agents in GRF

The difficulty of the LBF can be adjusted by the number of agents, the size of the grid world, the type of items, and the hard collaboration constraints. In this paper, we select 15×15-4p-5f as a supplement to the ablation experiment to verify whether the different components of our method are still valid in a more challenging collaborative setting. The 15×15-4p-5f scene indicates that 4 agents in a 15×15 grid world are involved in collecting 5 items.

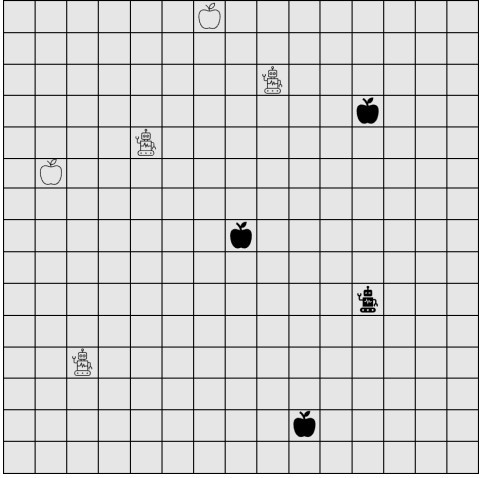

Figure 8: 15×15-4p-5f in LBF.

## B    CASE STUDY IN LBF

As discussed in Appendix A.3, the constraints from level and the sparse reward environment pose a greater challenge for agents to establish complementary and reliable collaborations. In order to verify the effectiveness of our proposed dynamic parameter sharing mechanism for the formation of complementary relationships between agents, we verify the performance of different parameter sharing mechanisms in 15×15-4p-5f, including single-layer dynamic parameter sharing (ours with only individual level), bi-layer dynamic parameter sharing (ours), full static parameter sharing (classic QMIX) and no parameter sharing (QMIX with non-sharing) at all.

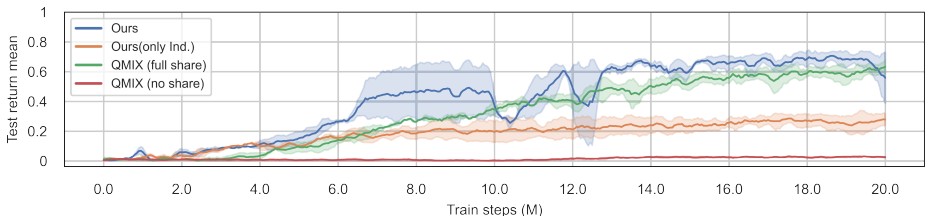

Figure 9: Comparison of different parameter sharing mechanisms in 15×15-4p-5f.

As shown in Figure 9, our method still yields the best return, demonstrating that our approach promotes more collaboration among agents. Although single-layer dynamic parameter sharing can promote agent collaboration, it can not always form complementary collaborative relationships. The result again confirms that forming complementary collaborations among agents requires a dynamic balance of personality and commonality, which is hidden in dynamic parameter sharing at both the individual and team levels.

## C   ABLATION STUDY IN GRF

As shown in Figure 8, we additionally verified the performance of the single-layer dynamic parameter sharing mechanism (only individual level) in two GRF scenarios.

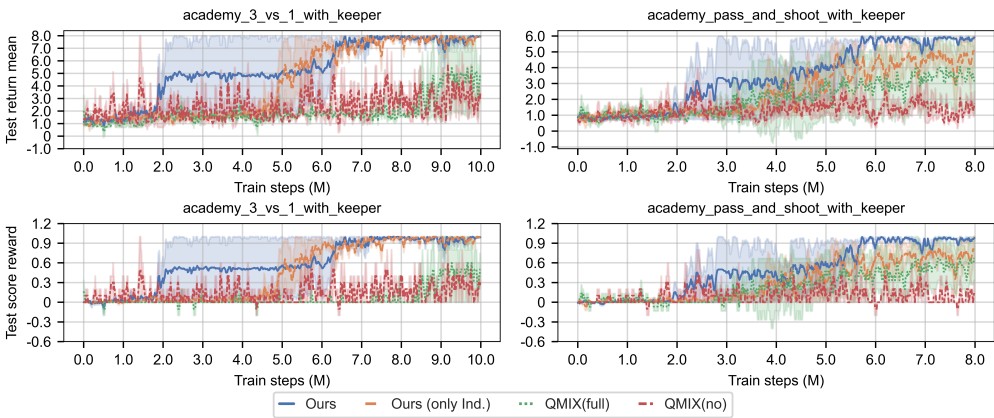

Figure 10: Performance comparison with baselines on Google Research Football.

Unlike the experimental results of SMAC and LBF, the algorithm of single-layer dynamic parameter sharing is better than the QMIX algorithm of full static parameter sharing and even can achieve the result of bi-level parameter sharing in $academy\_3\_vs\_1\_with\_keeper$. The experimental phenomena is because, in the GRF environment, agents have a more extensive range of actions, which makes them have a unique observation space than the other two environments, which proves that the full parameter is more effective because agents have similar observation spaces(Christianos et al., 2021).

We need to point out that the single-layer dynamic parameter sharing mechanism in the two scenarios is not as stable as our bi-level dynamic parameter sharing mechanism. This is because, on the one hand, agents are unstable due to changes in shared partners; on the other hand, agents that do not share parameters cannot establish complementary cooperation relationships, which again proves that our approach is correct in considering both individual and team levels.

## D  THE EFFECT OF THE NUMBER OF VIRTUAL ROLES

During the experiment, we found that setting different numbers of virtual roles would affect the performance of the entire algorithm. As shown in Figure 11, we show that in the SMAC environment, we only consider the impact of different numbers of virtual roles on the experimental results.

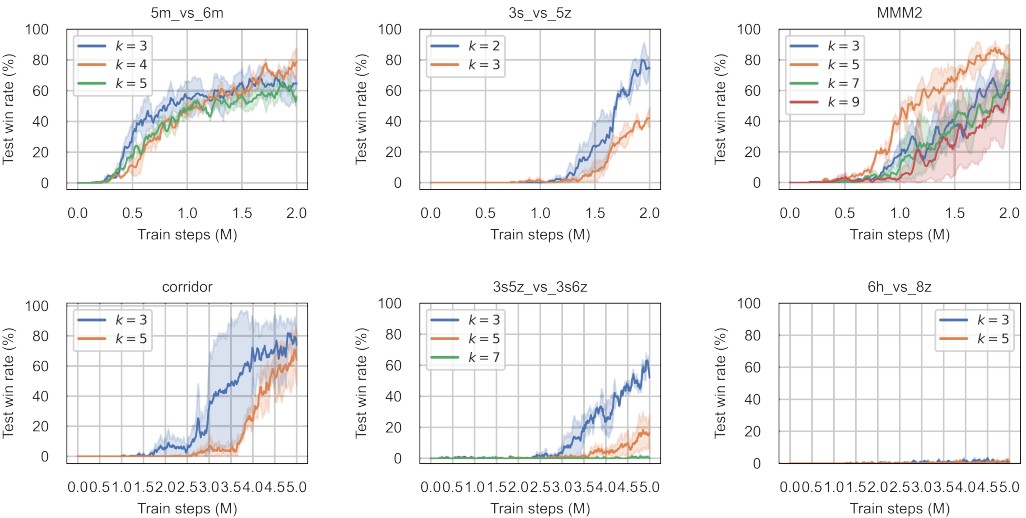

Figure 11: Compare the effects of different numbers of virtual roles (only individual level).

As we analyzed in the text, setting too many virtual roles will not improve the performance of the algorithm.

## E  FULL EXPERIMENTAL RESULTS IN SMAC

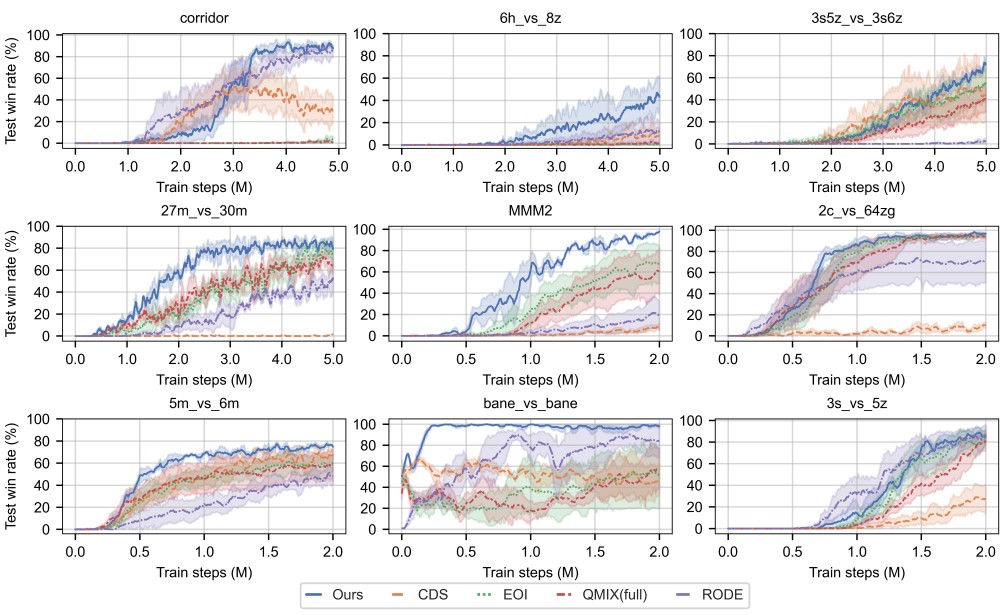

Figure 12: Performance comparison with baselines on SMAC.

## F    Hyperparameters Setting

In our method, we design a role selector and a group selector to guide agents in selecting appropriate partners for dynamic parameter sharing at the individual and team levels. The core of the role selector is a gated recurrent unit variational autoencoders (GRU-VAE). Besides the GRU unit, the encoder and decoder parts that makeup VAE are composed of simple linear layers and activation functions. The design of the group selector can be divided into a graph attention network with a 128-dimensional output state and other corresponding dimensions of the network layers (a linear layer and a GRU unit with a 64-dimensional hidden state). To expand, we use the softmax function to get the roles of agents at the individual level through the bottleneck of the GRU-VAE. For the grouping of agents at the team level, considering that our grouping task is an extra-designed reinforcement learning task, we use the same $\varepsilon - greedy$ method as agents' selection of actions.

We need to state that, in addition to the newly designed components of our method, we maintain as much agreement as possible with the baseline method QMIX on the structure, parameters, and optimization methods that make up the rest of our method, so as to fully demonstrate in ablation experiments that the real utility of the different components that make up our method comes entirely from our design.

## G    Experimental Details

For our method and baseline algorithms used, we tested $2M$ and $5M$ steps on all hard maps and all super hard maps of SMAC using five random seeds, respectively. Due to the use of the $\varepsilon - greedy$ method, we set $\varepsilon$ to be linearly annealed from $1.0$ to $0.05$ in $70K$ time steps in all hard maps and three super hard maps and kept constant during the rest of the training period. We set the annealing time for the remaining super hard map 6h_vs_8z to $500K$.

In GRF, we respectively tested $10M$ and $8M$ steps on $academy\_3\_vs\_1\_with\_keeper$ and $academy\_pass\_and\_shoot\_with\_keeper$ using three random seeds. We set $\varepsilon$ to be linearly annealed from $1.0$ to $0.05$ in $100K$ time steps in both scenarios. For part of the observable space we designed, it contains the location information of the agent itself, the position, direction and attribution information of the football, as well as the player information and opponent information within the visual range.

To test our method components' effectiveness in the $15 \times 15$-4p-5f scenario of the LBF environment, we set the experimental test step as $20M$ and the annealing time of $\varepsilon$ as $200K$. The setting of $\varepsilon$ remains the same in our method and baseline algorithms.

For the section on the virtual roles and groups setup in our method, in order to ensure that the collaborative relationship between agents is sufficiently complementary, we set the number of agents $n$, the number of virtual roles $k$, and the number of groups $m$ satisfying the relationship $n \geq k \geq m \geq 2$. Of course, as we obtained plots of the number of virtual roles and the win rate in our ablation experiments, the number of virtual roles could not be increased blindly. Blindly increasing the number of virtual roles not only brings a vast amount of parameters, but also makes complementary collaborations between agents impossible. In the SMAC and the LBF, the number of virtual roles we set is shown in Table 2. In response to agents' virtual roles and groups update periods, we have $T_V = e \cdot T_U$. To guarantee that the virtual roles and groups are updated normally in each episode, we require that the update periods of the virtual roles and the groups satisfies $T_U, T_V < \text{len}(episode)$. We set update periods for virtual roles and groups based on the average length of a episode. In general, we set the update periods of $5, 7, 8, 10$ or $14$ with $e = 1$ or $2$.

Our experiments are completed at Ubuntu 18.04 with a GPU of NVIDIA GTX 3090, a CPU of Intel i9-12900k and a memory size of 128G. In addition, the SMAC version is 4.10, the experimental version of $15 \times 15$-4p-5f in LBF is v2 and the version of the GRF is 2.10. We will open source my algorithm and experimental environments code on time.

Table 2: Number of virtual roles and groups in different maps.

| Environment Name | Map Name | Number of Roles | Number of Groups |
|---|---|---|---|
| SMAC | bane_vs_bane | 3 | 2 |
| | MMM2 | 5 | 2 |
| | 5m_vs_6m | 3 | 2 |
| | 3s_vs_5z | 2 | 2 |
| | 2c_vs_64zg | 2 | 2 |
| | 27m_vs_30m | 3 | 2 |
| | 6h_vs_8z | 3 | 2 |
| | corridor | 3 | 2 |
| | 3s5z_vs_3s6z | 3 | 2 |
| LBF | 15×15-40-5f | 2 | 2 |
| GRF | 3_vs_1_with_keeper | 2 | 2 |
| | pass_and_shoot_with_keeper | 2 | 2 |

