# OpenReview forum: "Bi-Level Dynamic Parameter Sharing among Individuals and Teams for Promoting Collaborations in Multi-Agent Reinforcement Learning"
_ICLR.cc/2023/Conference — Submitted to ICLR 2023_

### Official Review · Reviewer_diDZ · 2022-10-22

**Confidence:** 4
**Correctness:** 3
**Technical Novelty And Significance:** 2
**Empirical Novelty And Significance:** 2
**Recommendation:** 3

**Clarity, Quality, Novelty And Reproducibility:**

While the motivation for the work is clear, the motivation behind the decisions made throughout are not.
The work seems reasonably novel, though it does share similarities with RODE and other hierarchical MARL methods.
Regarding reproducibility, the architecture is quite complex and code has not been submitted, so it may be difficult to reproduce.

**Strength And Weaknesses:**

# Strengths
* The method addresses a challenging and important problem.
* The results seem to indicate that it is effective in improving upon standard baselines (e.g. QMIX).

# Weaknesses
* It's unclear what the VAE over "advantages" is representing, especially since the advantages are computed with agent utility values which are **not** Q-values in the traditional sense since they do not respect the Bellman equation. These agent-wise utilities are fed into a mixing network to produce a $Q_\text{tot}$ which is a true Q-value.
* The choice to select the argmax of the VAE latent as the role (Eqn 3) seems unmotivated.
* If I understand correctly, the group selector requires information from all agents (the set of all virtual roles). As such, the method does not fall within the decentralized execution paradigm as suggested by section 2.2, and this should be more explicitly stated.
* The group selector appears to be a hierarchical controller which operates over a dilated time scale, but in order to learn it we would need to have a separate mixing network for the group-selection actions. This is vaguely alluded to in the text, but is not shown in Figure 1. Furthermore, the Q-functions in Eqn 8 would have to be defined over the domain of group-selection actions, $\bf v$, but they are not which is especially confusing since the equation is maximizing over those.
* The results for RODE are not consistent with the literature. I haven't checked the others carefully, but those stood out since it thoroughly outperforms QMIX in the original paper.
* The ablations aren't explained clearly. More granular ablations are required to justify the design decisions (e.g. using attention, VAE, etc.)

# Questions
* What is meant by "advantage information can better represent the current state of agents than the observation information"? If anything these filter out information about the agents' observations to only that which is relevant to the global expected returns.

**Summary Of The Paper:**

This paper presents a method for dynamic parameter sharing over individual and group levels.
Results show an improvement over standard baselines on two challenging benchmarks (Google Research Football and StarCraft).

**Summary Of The Review:**

While the results are reasonably good, I came away from the paper unsure why most of the specific design decisions were made, and the experiments did not provide a significantly granular ablation study to justify those decisions. The paper could improve by providing both more intuitive explanations for the decisions as well as improved ablations.

---

### Official Review · Reviewer_CKA2 · 2022-10-23

**Confidence:** 4
**Correctness:** 2
**Technical Novelty And Significance:** 1
**Empirical Novelty And Significance:** 2
**Recommendation:** 3

**Clarity, Quality, Novelty And Reproducibility:**

The paper is not very well-written and hard for me to follow. Many descriptions about the proposed method are ambiguous or confusing to me (see my detailed comments below).

The proposed method does not look novel enough to me. Variational autoencoder and graph attention network have been widely used in existing works and the proposed method seems to be using them in a rather naive way. The idea of using the advantage sequence information to select individual roles for the agents seems to be new. But I do not quite understand the intuition behind it.

**Strength And Weaknesses:**

- Strengths:
	- The proposed method seems to achieve some good experimental results in different cooperative tasks.
- Weaknesses:
	- The proposed method is not very well motivated and the key insights/intuitions behind it are not very clear.
	- Many details of the proposed method are not clear to me, making it hard for me to judge the technical contributions.

**Summary Of The Paper:**

This paper proposes a bi-level dynamic parameter sharing mechanism (BDPS) to achieve better coordination among agents in cooperative multi-agent tasks. The key idea is rather than blindly sharing parameters among all agents, they share parameters among agents based on their roles and groups.  At the agent level, they determine the roles of different agents based on the advantage sequence information and agents with the same role share parameters. At the team level, they determine the groups of different roles using a graph attention network and agents within the same group share parameters.

**Summary Of The Review:**

I struggled with understanding the main method proposed in this paper. I found many descriptions about the proposed method confusing and hard to understand. Just to list a few examples here:
- "we give the time relationship for agents to update the groups and roles: T_V = e · T_U." Why the update periods of the group and individual roles follow this relationship? How do you decide the value of T_V  and T_U?
- "because the advantage information can better represent the current state of agents than the observation information." The advantage function basically describes how much better or worse an action selection is compared to the agent’s performance under current policy. Why it can better represent the current state of agents than the observation information? What is the intuition behind this?
- "use the \epsilon−greedy to select groups v_V for agents." It is not clear to me how do you really select groups for the agents (e.g., how to select groups when you are being *greedy*). Why is this a good way of selecting groups for agents (with probability \epsilon, you are just randomly choosing groups for agents)?
- "For the group selector V at the team level, we used the QMIX (Rashid et al., 2018) and RODE (Wang et al., 2021b) methods because we introduced an additional deep reinforcement learning task to generate groups for agents." This is not a good explanation of the motivation for using QMIX and RODE. It is also not clear to me how RODE is used in the proposed method.
- In Equation 8, how is $Q_{tot}^{V}$ generated? Through a mixing network? I cannot find it in the overall BDPS architecture (Figure 1). There is the global action value function $Q_{tot}$ in Equation 9, does this mean there should also be a $Q_{tot}^{U}$, which is never mentioned (I believe)?

In addition, I'm not fully convinced that the current way of combining agents of different roles into groups in BDPS is beneficial. Simply using $\epsilon$−greedy strategy to select groups for the agents seems random and naive to me. I do not see how randomly assigning groups sometimes can help boost the performance. Also, the result of the single-layer dynamic parameter sharing mechanism (only individual level) in one GRF scenarios shows that its final performance is very similar to that of BDPS. How does the single-layer dynamic parameter sharing mechanism perform in all SMAC maps tested?

---

### Official Review · Reviewer_n4Px · 2022-10-24

**Confidence:** 4
**Correctness:** 2
**Technical Novelty And Significance:** 3
**Empirical Novelty And Significance:** 2
**Recommendation:** 3

**Clarity, Quality, Novelty And Reproducibility:**

The clarity of this paper can be improved. (1) Figure 1 can be simplified and some irrelevant components should be removed to make it more clear. (2) Figure 2 is not clear. (3) The definition of roles and teams is missing.



**Strength And Weaknesses:**

**Strengths**

1. The problem this paper considers is very important. Parameter sharing could largely improve the efficiency of policy learning in MARL.

2. The literature review is sufficient. Previous works are discussed and compared empirically.
3. The motivation for this work is easy to understand. Previous works only considered single-level parameter sharing, and this paper novelly considers bi-level sharing.


**Weaknesses**

1. The proposed method is not sound enough. (1) In Definition 1, it seems that a role can only belong to one team, but why is that? And what are the definitions of roles and teams? (2) Individual-level parameter sharing depends on advantages, which are calculated using $Q$ values. What if these $Q$ values are not accurate at the beginning of training? Will the false role selection further hamper policy learning? (3) Advantages are computed as $A=\max{(Q)}-Q$, then agents with similar observations will always have the same role. What if two similar agents need to take different sub-tasks?
2. The quality of empirical evaluations needs to be improved. (1) Figure 2 is not clear. Do not use dotted lines for curves with similar performance. (2) In Figure 3, the performances of CDS and RODE in some tasks are very different from their original paper. (3) The ablation studies are not enough. More results are expected. What is the functionality of individual-level/team-level parameter sharing? How do the roles and teams evolve in an episode or in the process of training?
3. The results are not significant. This paper tested two tasks in Google Football, but the proposed method only outperforms others in one task.
4. This paper does not provide any discussions of limitations.

**Summary Of The Paper:**

This paper focuses on parameter-sharing issues in multi-agent reinforcement learning. The core idea of this paper is that previous methods of parameter sharing only consider the sharing between different agents, which is a single-level sharing. This work proposes a bi-level parameter-sharing mechanism that not only shares parameters among agents but also among teams. To achieve this, they defined roles based on the long-term cumulative advantages and share parameters of agents in the same team. Empirically, they evaluated the proposed method in SC2 and Google Footballs. And the results show some strengths over baselines.

**Summary Of The Review:**

The idea of bi-level parameter sharing is novel and interesting, but this paper can be further improved especially in terms of soundness, clarity and ablation studies.

---

### Official Review · Reviewer_UyCu · 2022-11-04

**Confidence:** 4
**Correctness:** 3
**Technical Novelty And Significance:** 2
**Empirical Novelty And Significance:** 2
**Recommendation:** 3

**Clarity, Quality, Novelty And Reproducibility:**

Both the writing and the quality of the paper should be further improved.

The code is not attached in the Appendix.



**Strength And Weaknesses:**

**Weaknesses**
  * The motivation is not clear enough. From the paper, I do not see why using the advantage function to represent the roles of agents is better and why incorporating a two-level selective parameter sharing is important.
  * The writing of the paper could be significantly improved. Lots of sentences are too long and many sentences are unclear, e.g., "we add additional implicit information $h_U^t$ from the individual role level to give full play to the positive impact of the individual identities on the team results."
  * Two important single-layer selective parameter sharing baselines ([1] and [2]) are mentioned in the introduction but are missing in the experiments.
  * The experimental evaluations in SMAC is not convincing. Recently, [3] and [4] have verified that the optimized QMIX (completely sharing the parameters among all agents) could achieve 100% win rates on all Easy, Hard and Super Hard scenarios of SMAC. Therefore, SMAC may not be a good testbed to validate the benefits of the dynamic parameter sharing mechanism. The authors could verify the algorithm in more complex environments.
  * minor:
    * Typos:
       * "And according to the role selector and the U group selector, choose the most suitable role and group for the agents V."
       * In equation (6), the local Q-function improperly takes the joint action **a** as input.

**Reference**
  * [1] Scaling multi-agent reinforcement learning with selective parameter sharing.
  * [2] A cooperative multi-agent reinforcement learning algorithm based on dynamic self-selection parameters sharing.
  * [3] Rethinking the implementation tricks and monotonicity constraint in cooperative multi-agent reinforcement learning
  * [4] API: Boosting Multi-Agent Reinforcement Learning via Agent-Permutation-Invariant Networks

**Summary Of The Paper:**

This paper proposes a Bi-level Dynamic Parameter Sharing mechanism (BDPS) in MARL. The core idea is assigning different agents to different roles based on the long-term cumulative advantages and grouping multiple roles to a set of teams via the Graph Attention Network. And the authors verify the proposed method in some typical MARL benchmarks.

**Summary Of The Review:**

This paper study an important and an interesting problem in MARL. But the quality of the paper should be significantly improved. So I recommend a reject based on the current version of the paper.

---

### Decision · Program_Chairs · 2023-01-20

**Decision:**

Reject

**Justification For Why Not Higher Score:**

Unanimous agreement among reviewers and no author rebuttal.

**Justification For Why Not Lower Score:**

N/A

**Metareview: Summary, Strengths And Weaknesses:**

The reviewers unanimously agree that this paper is not ready for publication. Therefore I recommend rejection.